# Properties of Fibre-Reinforced High-Strength Concrete with Nano-Silica and Silica Fume

Arash Karimipour [1,*], Mansour Ghalehnovi [2], Mahmoud Edalati [3] and Jorge de Brito [4,*]

1 Department of Civil Engineering, University of Texas at El Paso and the Member of Centre for Transportation Infrastructure System, El Paso, TX 79968, USA
2 Department of Civil Engineering, Ferdowsi University of Mashhad, Mashhad 91779-48974, Iran; Ghalehnovi@um.ac.ir
3 Department of Civil Engineering, Ilam University, Ilam 69311-69991, Iran; Edalati.mahmoud@ilam.ac.ir
4 Department of Civil Engineering, Architecture and Georresources, Instituto Superior Técnico, Universidade de Lisboa, 1649-004 Lisbon, Portugal
* Correspondence: akarimipour@miners.utep.edu (A.K.); jb@civil.ist.utl.pt (J.d.B.)

**Abstract:** This study intends to assess the influence of steel fibres (SF) and polypropylene fibres (PPF) on the hardened and fresh state properties of high-strength concrete (HSC). For this purpose, 99 concrete mixes were designed and applied. SF and PPF were used at six-volume replacement contents of 0%, 0.1%, 0.2%, 0.3%, 0.4% and 0.5%. Moreover, nano-silica (NS) was used at three contents, 0%, 1% and 2%, and silica fume powder (SP) was also used at three weight ratios (0%, 5% and 10%). The slump, compressive and tensile strength, elasticity modulus, water absorption and the electric resistivity of concrete specimens were examined. The results showed that using 1% NS and 10% SP together with 0.5% PPF improved the compressive strength of HSC by about 123%; however, the effect of SF on tensile strength is more significant and adding 0.5% SF with both 2% NS and 10% SP increased the tensile strength by 104%. Moreover, increasing the SF content reduces the electric resistivity while using PPF improves this property especially when 1% NS was employed, and it was enhanced by about 68% when 0.5% SF and 1% NS were utilized with 10% SP.

**Keywords:** polypropylene fibres; steel fibres; nano-silica; silica fume; high-strength concrete

## 1. Introduction

Cement is one of the most largely produced materials in the world. A high value of greenhouse gases is released into the environment within the process of manufacturing cement, affecting the health of humans, animals and plants [1]. The cement industry is one of the largest industrial manufacturers of carbon dioxide ($CO_2$), more than 5% of which 50% is released due to the chemical process and 40% from burning fuel. This industry, by producing around 7% of all $CO_2$ emissions, is the second-highest emitter of all industrial carbon dioxide [2]. Using waste materials is a useful way to reduce the amount of harmful emitted gases in industrial processes. In this area, many types of research have been carried out to study the influence of silica fume powder (SP) on the mechanical behaviour of concrete. Hemavathi et al. [3] employed SP and glass fibres to manufacture high-strength concrete (HSC). For this aim, SP replaced cement at 20% by weight. Moreover, 1% of glass fibres were used, and marble was utilized at three weight contents of 0%, 50% and 100%. In another study, Wu et al. [4] assessed the influence of SP (ranging from 0% to 25%) on the mechanical and rheological performance of HSC. Besides, 2% steel fibres (SF) were added to the concrete mixes. For this purpose, the compressive, flexural and tensile strengths of concrete were examined. The composite theory was also employed for numerical simulation. The results showed that the maximum compressive, flexural and tensile strengths could be obtained using 10% to 15% SP. On the other hand, the theoretical methods could predict the mechanical behaviour of HSC when SP was used

at up to 20%. Therefore, the predicted-to-experimental tensile and flexural strength ratios were in the range of 1.1 to 0.9. Guo et al. [5] examined the mechanical performance of self-compacting concrete (SCC) containing recycled aggregate, fly ash, SP and slag. The purpose of this investigation was to determine the optimal value of recycled aggregate paying attention to the contents of fly ash, SP and slag. For this aim, 23 concrete mixes were employed, and the mechanical and durability characteristics of SCC were evaluated. The experimental results demonstrated that using an optimal combination of SP, fly ash and slag significantly enhanced the mechanical properties of recycled aggregate SCC. In 2017, Motahari et al. [6] studied the impact of SP on the mechanical performance of concrete. For this aim, different contents of SP were used as partial replacement of cement to improve the durability properties of concrete. The results showed that using SP resulted in increasing the compressive strength and electric resistivity of concrete and reducing the permeability of concrete.

In another investigation, Xie et al. [7] assessed the bending performance of recycled aggregate concrete beams containing SP and polypropylene fibres (PPF). Therefore, different specimens were manufactured and tested in a three-point flexural setup. The compressive strength, modulus of elasticity and failure mode of recycled aggregates concrete beams were the aims of this research. Based on the conclusions of this research, using both SP and PPF led to improving the compressive and flexural performance of recycled aggregate concrete. Gökçe et al. [8] studied the impact of fly ash and SP on the hardened state properties of foam concrete at different curing ages. For this purpose, 45 concrete mixes were designed. Fly ash and SP were employed at three different contents (0%, 10% and 20% by weight). Foam was used at 0%, 31% and 47%. Specimens were tested under hydraulic jacks at 7 and 28 days, and water absorption, compressive strength and thermal conductivity were measured. It was demonstrated that SP improves the value of compressive strength more than fly ash. Amin et al. [9] examined the 90-day concrete compressive strength by replacing cement with 30% marble and 0%, 5% and 10% SP, obtaining reductions of up to 20%, 60% and 45% respectively. In 2019, Zareei et al. [10] used nano-silica (NS) and slag to manufacture sustainable HSC. For this aim, 2% of cement was replaced with NS, and slag was utilized at four replacement ratios of 25%, 50%, 75% and 100% instead of sand. It was shown that increasing the value of NS and slag led to improving the strength and durability of concrete; however, workability dropped. In another research, Gupta and Kumar [11] evaluated the influence of NS and coir fibre on the mechanical performance of concrete. Therefore, fibres were utilized at 0.25%, 0.5%, and 0.75% by weight of fine aggregates and 2% and 3% NS with 15% fly ash replaced cement. According to the results, the optimal contents of fibres and NS in terms of abrasion resistance were 0.25% and 3%, respectively. In 2018, Olivier et al. [12] considered the influence of NS, super absorbent polymers and synthetic fibres on the shrinkage cracking of concrete. It was shown that the use of NS in concrete should be carefully examined. Moreover, using both NS and super absorbent polymers considerably improved the shrinkage cracking. Zeng et al. [13] employed NS to improve the corrosion resistance of recycled aggregate concrete. Therefore, both the fresh and hardened state properties of concrete were examined. The improvement of the corrosion resistance of concrete when NS was utilized was the main conclusion of this study.

Çevik et al. [14] used NS and fly ash to improve the mechanical characteristics and chemical durability of geopolymer concrete. Four types of geopolymer and ordinary concrete mixes were subjected to sulphuric acid, magnesium sulphate and seawater solutions with concentrations of 5%, 5% and 3.5%, respectively. To evaluate the mechanical performance of the specimens, compressive, splitting tensile and flexural strength tests were carried out. Based on the experiments, NS enhances both durability and residual mechanical strength because of low porosity and higher density. Golafshani and Behnood [15] proposed an optimal value of NS to optimize the mechanical properties of concrete. In addition, a high-fitting formula was proposed to forecast the compressive strength of concrete and the optimal SP content. In the numerical simulation process, the contents of

cement, water, SP and aggregates, the maximum size of aggregate and curing age were the variables under analysis. It was demonstrated that the proposed formula could accurately predict the optimal value of SP in concrete. Mehta et al. [16] published a review article in the area of the influence of SP and waste glass on the mechanical performance of concrete. In addition, a microstructural analysis was employed. Khan and Ali [17] analysed the behaviour of concrete with fly ash, SP and coconut fibres. 15% SP replaced cement and fly ash was employed at four contents by weight: 0%, 5%, 10% and 15%. Besides, 2% of coconut fibres were added to the concrete mixes. The compressive strength, splitting tensile strength, stress-strain relationship, load-deflection and load-time curves of specimens were assessed in this investigation. According to the results, using both coconut fibres and 10% of fly ash significantly improved the mechanical properties of concrete.

In 2019, Sasanipour et al. [18] studied the impact of SP on the durability of recycled aggregates SCC. Specimens were manufactured in three families. In the first and second ones, coarse recycled aggregates replaced natural aggregates at four contents: 25%, 50%, 75% and 100%, with and without SP respectively. In the third family, 25% of fine recycled aggregates replaced fine natural aggregates. Both the fresh and hardened state properties of SCC were examined. SP significantly improved the fresh state properties of SCC. Conversely, increasing the recycled aggregates' content resulted in reducing the electric resistivity. SP also controlled the temperature of the solutions used during the test. Esfandiari and Loghmani [19] examined the impact of perlite powder and SP on the compressive strength and microstructural properties of SCC with a lime-cement binder. For this purpose, SP and perlite powder were employed at 0%, 5%, 10% 15% and 20%. It was demonstrated that the water absorption, dry density and compressive strength increased by increasing the SP content.

## 2. Research Significance

A review of the literature showed that considerable research work has been carried out to evaluate the influence of SP and NS separately on the mechanical and durability performance of concrete. It showed that the contents of SP and NS should be limited and using high-contest of these materials leads to reducing the mechanical properties of concrete [12–15]. On the other hand, previous investigations demonstrated that using different types of fibres plays a positive role on the mechanical behaviour of concrete especially in terms of compressive, tensile and flexural strength [20–32]. Guo et al. [33] evaluated the effect of PPF and SF on the static and dynamic characteristics of HSC. They found that adding SF resulted in brittle failure in the indirect tensile strength test, while the simultaneous incorporation of SF and PPF led to ductile failure. Besides, the resistance was substantially enhanced with increasing PPF and SF content due to the bridging role of the fibres and their higher tensile strength in comparison with the tensile strength of the concrete matrix. Mo et al. [34] investigated the mechanical properties of PPF-reinforced concrete with rubber aggregates. They showed that adding PPF led to reducing the dilapidation rate of the dynamic bending stiffness. Rashid [35] used SF and PPF to improve the durability performance of concrete under natural weathering. The specimens were exposed to natural conditions for 360 days. Additionally, the compressive strength, water absorption and porosity specimens containing various PPF content were studied. It was found that PPF significantly improved the durability and mechanical characteristics of the specimens. Therefore, finding a way to increase the contents of SP and NS in order to improve the mechanical and durability performance of concrete by combining these materials was the main aim of this study. Moreover, PPF and SF were used to prevent the reduction effect of using a high content of SP and NS on the properties of concrete.

## 3. Materials and Specimens' Specifications

In this research, ordinary Portland cement (type I) was used to manufacture concrete according to ASTM C150/C150M [36]. The physical and chemical characteristics of cement are represented in Tables 1 and 2. Moreover, the mechanical properties of aggregates are

illustrated in Table 3, according to the requirement of ASTM D2419 [37]. The grading curves of the aggregates are also shown in Figure 1 and Table 4, as per ASTM C33 [38].

**Table 1.** Physical properties of cement.

| Final Setting Time (min) | Initial Setting Time (min) | Specific Surface (cm$^2$/g) | Specific Gravity (g/cm$^3$) | Autoclave Expansion (%) |
|---|---|---|---|---|
| 200 | 150 | 3140 | 3.11 | 0.08 |

**Table 2.** Chemical properties of cement.

| SiO$_2$ (%) | Al$_2$O$_3$ (%) | Fe$_2$O$_3$ (%) | CaO (%) | MgO (%) | SO$_3$ (%) | Na$_2$O (%) | K$_2$O (%) | C$_3$S (%) | C$_2$S (%) | C$_3$A (%) | C$_4$AF (%) | Free CaO | Loss on Ignition | Insoluble Residue |
|---|---|---|---|---|---|---|---|---|---|---|---|---|---|---|
| 16.78 | 4.92 | 3.43 | 6.41 | 2.26 | 2.41 | 0.41 | 0.62 | 60.08 | 1.13 | 0.43 | 0.97 | 0.15 | 2.35 | 0.51 |

**Table 3.** Properties of the aggregates.

| Fine Aggregates | | | |
|---|---|---|---|
| Fineness Modulus | Specific Gravity (g/cm$^3$) | Water Absorption (%) | Maximum Grain Size (mm) |
| 2.7 | 2.74 | 1.73 | 4.85 |
| **Coarse Aggregates** | | | |
| Specific Gravity (g/cm$^3$) | Water Absorption (%) | Maximum Nominal Size (mm) | |
| 2.72 | 0.49 | 12.6 | |

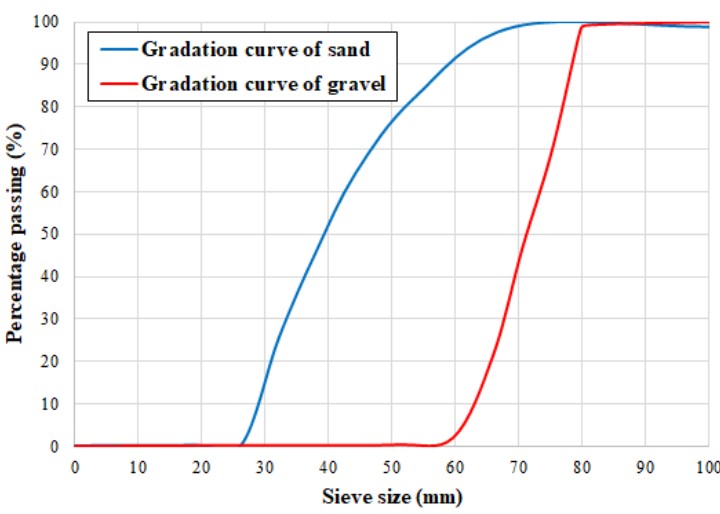

**Figure 1.** Gradation curve of aggregates.

**Table 4.** Grading of the aggregates.

| Parameter | Type of Aggregate | Sieve Size | | | | | | | | |
|---|---|---|---|---|---|---|---|---|---|---|
| | | 3/4 in (19 mm) | 1/2 in (12.5 mm) | 3/8 in (9.5 mm) | No. 4 (4.75 mm) | No. 8 (2.36 mm) | No. 16 (1.18 mm) | No. 30 (0.6 mm) | No. 50 (0.3 mm) | No. 100 (0.15 mm) |
| Passing ratio (%) | Coarse aggregate | 100 | 100 | 52.4 | 2.4 | 0.9 | - | - | - | - |
| | Fine aggregate | - | - | 100 | 98.2 | 80.4 | 70.1 | 52.2 | 28.1 | 6.4 |

To manufacture fibre concrete, SF and PPF were employed as illustrated in Figure 2. The properties of fibres are also presented in Table 5. In this study, the fibres were added

to the mix at three-volume contents of 0%, 1% and 2%. NS and SP were selected as the amorphous kind, F-110 grade A102, as a replacement for cement, respectively, as shown in Figure 3. Moreover, the physical and chemical properties of SP and NS are represented in Tables 6 and 7.

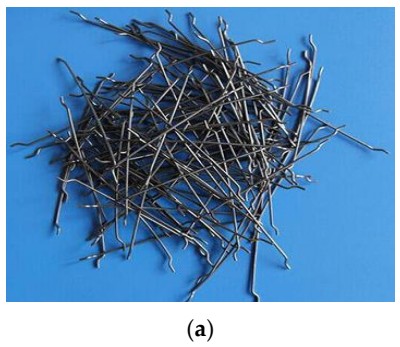

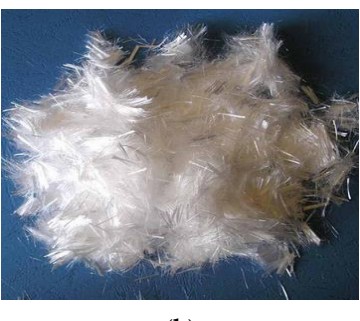

(**a**)　　　　　　　　　　　　　　　　　　　(**b**)

**Figure 2.** Fibres (**a**) SF (**b**) PPF.

**Table 5.** Properties of the fibres.

| Fibres | Tensile Strength (GPa) | Modulus of Elasticity (GPa) | Failure Strain (%) | Special Weight (kg/m$^3$) |
|---|---|---|---|---|
| SF | 200 | 2.0 | 3.0 | 2400 |
| PPF | 250 | 2.8 | 3.4 | 900 |

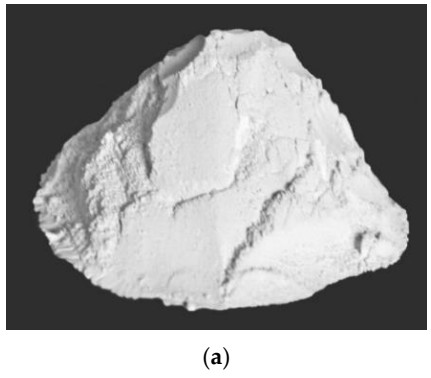

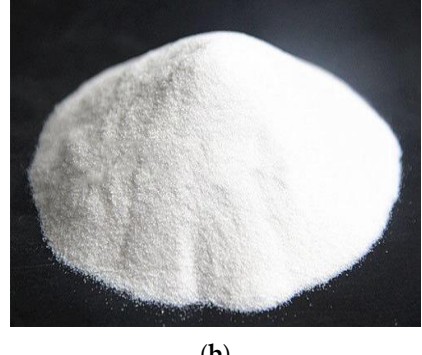

(**a**)　　　　　　　　　　　　　　　　　　　(**b**)

**Figure 3.** (**a**) SP and (**b**) NS.

**Table 6.** Chemical components of SP and NS.

| Material | SiO$_2$ (%) | Al$_2$O$_3$ (%) | Fe$_2$O$_3$ (%) | CaO (%) | MgO (%) | Na$_2$O (%) | SO$_3$ (%) | TiO$_2$ (%) | K$_2$O (%) | K$_2$O$_5$ (%) | ZnO (%) | CuO (%) |
|---|---|---|---|---|---|---|---|---|---|---|---|---|
| NS | $\geq 95$ | 3.481 | 0.285 | 0.387 | 0.05 | 0.325 | 0.175 | 0.058 | 0.075 | 0.125 | 0.015 | 0.020 |
| SP | 92 | 3.0 | 1.35 | 0.25 | 0.9 | 0.5 | - | - | 1 | - | - | - |

**Table 7.** Physical properties of SP and NS.

| Material | Loss on Ignition (%) | Loss on Drying (%) | Moisture (%) | Specific Gravity (g/cm$^3$) | Specific Surface (m$^2$/g) | Bulk Density (kg/m$^3$) | Mean Particle Size (mm) | PH |
|---|---|---|---|---|---|---|---|---|
| NS | - | < 5 | - | 2.55 | 185 | 50 | 20–30 | 5–6 |
| SP | 0.3–2.5 | - | 0.01–0.04 | 2 | 15 | 300–500 | 229 | 6.8–8 |

A high-performance superplasticizer was added until the fibres were uniformly distributed in the concrete matrix. The total water/cement ratio of all samples was kept constant at 0.35. To assess the compressive and tensile strengths of each concrete mix, six cylinders with a diameter of 150 mm and a height of 300 mm were manufactured and tested under a hydraulic jack [39–41]. The average compressive and tensile strength of three of those cylinders was considered as the specimens' strength. The mix proportions of specimens are also seen in Table 8. According to Table 8, concrete mixes were categorized into three general groups based on fibres content: group 1 without fibres, group 2 includes specimens containing SF, and group 3 includes specimens produced by PPF. In addition, groups 2 and 3 are divided into 5 sub-groups based on fibres contents: 0.1%, 0.2%, 0.3%, 0.4% and 0.5%. In addition, SF, PPF, SP and NS in Table 8 indicates the steel fibres, polypropylene fibres, silica fume powder and nano-silica contents, respectively.

**Table 8.** Concrete mixes design.

| Group | Sub-Group | Designation | Water | Cement | NS | SP | Fine Aggregate | Coarse Aggregate | Fibre (%) | | Superplasticizer (%) |
|---|---|---|---|---|---|---|---|---|---|---|---|
| | | | (kg/m$^3$) | | | | | | SF | PPF | |
| 1 | I | SP0-NS0 | 161.2 | 520 | 0 | 0 | 804.9 | 914.3 | 0 | 0 | 0.6 |
| | | SP0-NS1 | 161.2 | 514.8 | 5.2 | 0 | 804.9 | 914.3 | 0 | 0 | 0.88 |
| | | SP0-NS2 | 161.2 | 509.6 | 10.4 | 0 | 804.9 | 914.3 | 0 | 0 | 1.02 |
| | | SP5-NS0 | 161.2 | 494 | 0 | 26 | 804.9 | 914.3 | 0 | 0 | 1.16 |
| | | SP5-NS1 | 161.2 | 488.8 | 5.2 | 26 | 804.9 | 914.3 | 0 | 0 | 1.3 |
| | | SP5-NS2 | 161.2 | 483.6 | 10.4 | 26 | 804.9 | 914.3 | 0 | 0 | 0.825 |
| | | SP10-NS0 | 161.2 | 468 | 0 | 52 | 804.9 | 914.3 | 0 | 0 | 1.05 |
| | | SP10-NS1 | 161.2 | 462.8 | 5.2 | 52 | 804.9 | 914.3 | 0 | 0 | 1.275 |
| | | SP10-NS2 | 161.2 | 457.6 | 10.4 | 52 | 804.9 | 914.3 | 0 | 0 | 1.5 |
| 2 | I | SP0-NS0-SF0.1 | 161.2 | 520 | 0 | 0 | 804.9 | 914.3 | 0.1 | 0 | 0.6 |
| | | SP0-NS1-SF0.1 | 161.2 | 514.8 | 5.2 | 0 | 804.9 | 914.3 | 0.1 | 0 | 0.88 |
| | | SP0-NS2-SF0.1 | 161.2 | 509.6 | 10.4 | 0 | 804.9 | 914.3 | 0.1 | 0 | 1.02 |
| | | SP5-NS0-SF0.1 | 161.2 | 494 | 0 | 26 | 804.9 | 914.3 | 0.1 | 0 | 1.16 |
| | | SP5-NS1-SF0.1 | 161.2 | 488.8 | 5.2 | 26 | 804.9 | 914.3 | 0.1 | 0 | 1.3 |
| | | SP5-NS2-SF0.1 | 161.2 | 483.6 | 10.4 | 26 | 804.9 | 914.3 | 0.1 | 0 | 0.825 |
| | | SP10-NS0-SF0.1 | 161.2 | 468 | 0 | 52 | 804.9 | 914.3 | 0.1 | 0 | 1.05 |
| | | SP10-NS1-SF0.1 | 161.2 | 462.8 | 5.2 | 52 | 804.9 | 914.3 | 0.1 | 0 | 1.275 |
| | | SP10-NS2-SF0.1 | 161.2 | 457.6 | 10.4 | 52 | 804.9 | 914.3 | 0.1 | 0 | 1.5 |
| | II | SP0-NS0-SF0.2 | 161.2 | 520 | 0 | 0 | 804.9 | 914.3 | 0.2 | 0 | 0.6 |
| | | SP0-NS1-SF0.2 | 161.2 | 514.8 | 5.2 | 0 | 804.9 | 914.3 | 0.2 | 0 | 0.88 |
| | | SP0-NS2-SF0.2 | 161.2 | 509.6 | 10.4 | 0 | 804.9 | 914.3 | 0.2 | 0 | 1.02 |
| | | SP5-NS0-SF0.2 | 161.2 | 494 | 0 | 26 | 804.9 | 914.3 | 0.2 | 0 | 1.16 |
| | | SP5-NS1-SF0.2 | 161.2 | 488.8 | 5.2 | 26 | 804.9 | 914.3 | 0.2 | 0 | 1.3 |
| | | SP5-NS2-SF0.2 | 161.2 | 483.6 | 10.4 | 26 | 804.9 | 914.3 | 0.2 | 0 | 0.825 |
| | | SP10-NS0-SF0.2 | 161.2 | 468 | 0 | 52 | 804.9 | 914.3 | 0.2 | 0 | 1.05 |
| | | SP10-NS1-SF0.2 | 161.2 | 462.8 | 5.2 | 52 | 804.9 | 914.3 | 0.2 | 0 | 1.275 |
| | | SP10-NS2-SF0.2 | 161.2 | 457.6 | 10.4 | 52 | 804.9 | 914.3 | 0.2 | 0 | 1.5 |
| | III | SP0-NS0-SF0.3 | 161.2 | 520 | 0 | 0 | 804.9 | 914.3 | 0.3 | 0 | 0.6 |
| | | SP0-NS1-SF0.3 | 161.2 | 514.8 | 5.2 | 0 | 804.9 | 914.3 | 0.3 | 0 | 0.88 |
| | | SP0-NS2-SF0.3 | 161.2 | 509.6 | 10.4 | 0 | 804.9 | 914.3 | 0.3 | 0 | 1.02 |
| | | SP5-NS0-SF0.3 | 161.2 | 494 | 0 | 26 | 804.9 | 914.3 | 0.3 | 0 | 1.16 |
| | | SP5-NS1-SF0.3 | 161.2 | 488.8 | 5.2 | 26 | 804.9 | 914.3 | 0.3 | 0 | 1.3 |
| | | SP5-NS2-SF0.3 | 161.2 | 483.6 | 10.4 | 26 | 804.9 | 914.3 | 0.3 | 0 | 0.825 |
| | | SP10-NS0-SF0.3 | 161.2 | 468 | 0 | 52 | 804.9 | 914.3 | 0.3 | 0 | 1.05 |
| | | SP10-NS1-SF0.3 | 161.2 | 462.8 | 5.2 | 52 | 804.9 | 914.3 | 0.3 | 0 | 1.275 |
| | | SP10-NS2-SF0.3 | 161.2 | 457.6 | 10.4 | 52 | 804.9 | 914.3 | 0.3 | 0 | 1.5 |
| | IV | SP0-NS0-SF0.4 | 161.2 | 520 | 0 | 0 | 804.9 | 914.3 | 0.4 | 0 | 0.6 |
| | | SP0-NS1-SF0.4 | 161.2 | 514.8 | 5.2 | 0 | 804.9 | 914.3 | 0.4 | 0 | 0.88 |
| | | SP0-NS2-SF0.4 | 161.2 | 509.6 | 10.4 | 0 | 804.9 | 914.3 | 0.4 | 0 | 1.02 |
| | | SP5-NS0-SF0.4 | 161.2 | 494 | 0 | 26 | 804.9 | 914.3 | 0.4 | 0 | 1.16 |
| | | SP5-NS1-SF0.4 | 161.2 | 488.8 | 5.2 | 26 | 804.9 | 914.3 | 0.4 | 0 | 1.3 |
| | | SP5-NS2-SF0.4 | 161.2 | 483.6 | 10.4 | 26 | 804.9 | 914.3 | 0.4 | 0 | 0.825 |
| | | SP10-NS0-SF0.4 | 161.2 | 468 | 0 | 52 | 804.9 | 914.3 | 0.4 | 0 | 1.05 |
| | | SP10-NS1-SF0.4 | 161.2 | 462.8 | 5.2 | 52 | 804.9 | 914.3 | 0.4 | 0 | 1.275 |
| | | SP10-NS2-SF0.4 | 161.2 | 457.6 | 10.4 | 52 | 804.9 | 914.3 | 0.4 | 0 | 1.5 |

**Table 8.** *Cont.*

| Group | Sub-Group | Designation | Water | Cement | NS | SP | Fine Aggregate | Coarse Aggregate | Fibre (%) SF | Fibre (%) PPF | Superplasticizer (%) |
|---|---|---|---|---|---|---|---|---|---|---|---|
| | | | (kg/m$^3$) | | | | | | | | |
| | V | SP0-NS0-SF0.5 | 161.2 | 520 | 0 | 0 | 804.9 | 914.3 | 0.5 | 0 | 0.6 |
| | | SP0-NS1-SF0.5 | 161.2 | 514.8 | 5.2 | 0 | 804.9 | 914.3 | 0.5 | 0 | 0.88 |
| | | SP0-NS2-SF0.5 | 161.2 | 509.6 | 10.4 | 0 | 804.9 | 914.3 | 0.5 | 0 | 1.02 |
| | | SP5-NS0-SF0.5 | 161.2 | 494 | 0 | 26 | 804.9 | 914.3 | 0.5 | 0 | 1.16 |
| | | SP5-NS1-SF0.5 | 161.2 | 488.8 | 5.2 | 26 | 804.9 | 914.3 | 0.5 | 0 | 1.3 |
| | | SP5-NS2-SF0.5 | 161.2 | 483.6 | 10.4 | 26 | 804.9 | 914.3 | 0.5 | 0 | 0.825 |
| | | SP10-NS0-SF0.5 | 161.2 | 468 | 0 | 52 | 804.9 | 914.3 | 0.5 | 0 | 1.05 |
| | | SP10-NS1-SF0.5 | 161.2 | 462.8 | 5.2 | 52 | 804.9 | 914.3 | 0.5 | 0 | 1.275 |
| | | SP10-NS2-SF0.5 | 161.2 | 457.6 | 10.4 | 52 | 804.9 | 914.3 | 0.5 | 0 | 1.5 |
| | I | SP0-NS0-PPF0.1 | 161.2 | 520 | 0 | 0 | 804.9 | 914.3 | 0 | 0.1 | 0.6 |
| | | SP0-NS1-PPF0.1 | 161.2 | 514.8 | 5.2 | 0 | 804.9 | 914.3 | 0 | 0.1 | 0.88 |
| | | SP0-NS2-PPF0.1 | 161.2 | 509.6 | 10.4 | 0 | 804.9 | 914.3 | 0 | 0.1 | 1.02 |
| | | SP5-NS0-PPF0.1 | 161.2 | 494 | 0 | 26 | 804.9 | 914.3 | 0 | 0.1 | 1.16 |
| | | SP5-NS1-PPF0.1 | 161.2 | 488.8 | 5.2 | 26 | 804.9 | 914.3 | 0 | 0.1 | 1.3 |
| | | SP5-NS2-PPF0.1 | 161.2 | 483.6 | 10.4 | 26 | 804.9 | 914.3 | 0 | 0.1 | 0.825 |
| | | SP10-NS0-PPF0.1 | 161.2 | 468 | 0 | 52 | 804.9 | 914.3 | 0 | 0.1 | 1.05 |
| | | SP10-NS1-PPF0.1 | 161.2 | 462.8 | 5.2 | 52 | 804.9 | 914.3 | 0 | 0.1 | 1.275 |
| | | SP10-NS2-PPF0.1 | 161.2 | 457.6 | 10.4 | 52 | 804.9 | 914.3 | 0 | 0.1 | 1.5 |
| | II | SP0-NS0-PPF0.2 | 161.2 | 520 | 0 | 0 | 804.9 | 914.3 | 0 | 0.2 | 0.6 |
| | | SP0-NS1-PPF0.2 | 161.2 | 514.8 | 5.2 | 0 | 804.9 | 914.3 | 0 | 0.2 | 0.88 |
| | | SP0-NS2-PPF0.2 | 161.2 | 509.6 | 10.4 | 0 | 804.9 | 914.3 | 0 | 0.2 | 1.02 |
| | | SP5-NS0-PPF0.2 | 161.2 | 494 | 0 | 26 | 804.9 | 914.3 | 0 | 0.2 | 1.16 |
| | | SP5-NS1-PPF0.2 | 161.2 | 488.8 | 5.2 | 26 | 804.9 | 914.3 | 0 | 0.2 | 1.3 |
| | | SP5-NS2-PPF0.2 | 161.2 | 483.6 | 10.4 | 26 | 804.9 | 914.3 | 0 | 0.2 | 0.825 |
| | | SP10-NS0-PPF0.2 | 161.2 | 468 | 0 | 52 | 804.9 | 914.3 | 0 | 0.2 | 1.05 |
| | | SP10-NS1-PPF0.2 | 161.2 | 462.8 | 5.2 | 52 | 804.9 | 914.3 | 0 | 0.2 | 1.275 |
| | | SP10-NS2-PPF0.2 | 161.2 | 457.6 | 10.4 | 52 | 804.9 | 914.3 | 0 | 0.2 | 1.5 |
| 3 | III | SP0-NS0-PPF0.3 | 161.2 | 520 | 0 | 0 | 804.9 | 914.3 | 0 | 0.3 | 0.6 |
| | | SP0-NS1-PPF0.3 | 161.2 | 514.8 | 5.2 | 0 | 804.9 | 914.3 | 0 | 0.3 | 0.88 |
| | | SP0-NS2-PPF0.3 | 161.2 | 509.6 | 10.4 | 0 | 804.9 | 914.3 | 0 | 0.3 | 1.02 |
| | | SP5-NS0-PPF0.3 | 161.2 | 494 | 0 | 26 | 804.9 | 914.3 | 0 | 0.3 | 1.16 |
| | | SP5-NS1-PPF0.3 | 161.2 | 488.8 | 5.2 | 26 | 804.9 | 914.3 | 0 | 0.3 | 1.3 |
| | | SP5-NS2-PPF0.3 | 161.2 | 483.6 | 10.4 | 26 | 804.9 | 914.3 | 0 | 0.3 | 0.825 |
| | | SP10-NS0-PPF0.3 | 161.2 | 468 | 0 | 52 | 804.9 | 914.3 | 0 | 0.3 | 1.05 |
| | | SP10-NS1-PPF0.3 | 161.2 | 462.8 | 5.2 | 52 | 804.9 | 914.3 | 0 | 0.3 | 1.275 |
| | | SP10-NS2-PPF0.3 | 161.2 | 457.6 | 10.4 | 52 | 804.9 | 914.3 | 0 | 0.3 | 1.5 |
| | IV | SP0-NS0-PPF0.4 | 161.2 | 520 | 0 | 0 | 804.9 | 914.3 | 0 | 0.4 | 0.6 |
| | | SP0-NS1-PPF0.4 | 161.2 | 514.8 | 5.2 | 0 | 804.9 | 914.3 | 0 | 0.4 | 0.88 |
| | | SP0-NS2-PPF0.4 | 161.2 | 509.6 | 10.4 | 0 | 804.9 | 914.3 | 0 | 0.4 | 1.02 |
| | | SP5-NS0-PPF0.4 | 161.2 | 494 | 0 | 26 | 804.9 | 914.3 | 0 | 0.4 | 1.16 |
| | | SP5-NS1-PPF0.4 | 161.2 | 488.8 | 5.2 | 26 | 804.9 | 914.3 | 0 | 0.4 | 1.3 |
| | | SP5-NS2-PPF0.4 | 161.2 | 483.6 | 10.4 | 26 | 804.9 | 914.3 | 0 | 0.4 | 0.825 |
| | | SP10-NS0-PPF0.4 | 161.2 | 468 | 0 | 52 | 804.9 | 914.3 | 0 | 0.4 | 1.05 |
| | | SP10-NS1-PPF0.4 | 161.2 | 462.8 | 5.2 | 52 | 804.9 | 914.3 | 0 | 0.4 | 1.275 |
| | | SP10-NS2-PPF0.4 | 161.2 | 457.6 | 10.4 | 52 | 804.9 | 914.3 | 0 | 0.4 | 1.5 |
| | V | SP0-NS0-PPF0.5 | 161.2 | 520 | 0 | 0 | 804.9 | 914.3 | 0 | 0.5 | 0.6 |
| | | SP0-NS1-PPF0.5 | 161.2 | 514.8 | 5.2 | 0 | 804.9 | 914.3 | 0 | 0.5 | 0.88 |
| | | SP0-NS2-PPF0.5 | 161.2 | 509.6 | 10.4 | 0 | 804.9 | 914.3 | 0 | 0.5 | 1.02 |
| | | SP5-NS0-PPF0.5 | 161.2 | 494 | 0 | 26 | 804.9 | 914.3 | 0 | 0.5 | 1.16 |
| | | SP5-NS1-PPF0.5 | 161.2 | 488.8 | 5.2 | 26 | 804.9 | 914.3 | 0 | 0.5 | 1.3 |
| | | SP5-NS2-PPF0.5 | 161.2 | 483.6 | 10.4 | 26 | 804.9 | 914.3 | 0 | 0.5 | 0.825 |
| | | SP10-NS0-PPF0.5 | 161.2 | 468 | 0 | 52 | 804.9 | 914.3 | 0 | 0.5 | 1.05 |
| | | SP10-NS1-PPF0.5 | 161.2 | 462.8 | 5.2 | 52 | 804.9 | 914.3 | 0 | 0.5 | 1.275 |
| | | SP10-NS2-PPF0.5 | 161.2 | 457.6 | 10.4 | 52 | 804.9 | 914.3 | 0 | 0.5 | 1.5 |

Regarding Table 8, 99 mixes were produced. Then, six tests were carried out: slump, compressive and splitting tensile strengths, modulus of elasticity, water absorption and electrical resistivity. In each test, three specimens were tested for each mix and the average of three samples was considered. Therefore, 1782 specimens were manufactured and tested at 28 days of curing age.

## 4. Methods

In this section, the methodology of performed tests in this study is discussed, as below:

### 4.1. Slump

To measure the properties of the fresh state of HSC, slump flow was utilized. This test was performed according to ASTM C33 [38] to determine the workability of concrete. To that end, a circular truncated cone of 100 mm, 200 mm and 300 mm high, bottom and top diameters was filled with 3-layer fresh concrete and compacted by 25 strikes each. Slowly the mould was then raised and the height of the concrete settlement was deemed to be the downside.

### 4.2. Compressive Strength

To assess the hardened state performance of HSC, the effect of NS, SP, SF and PPF on the compressive strength of HSC was examined according to ASTM C39 [42], For this aim, three 150 mm × 300 mm cylindrical specimens were subjected to a hydraulic jack, and the average of three samples was considered.

### 4.3. Splitting Tensile Strength

To measure, the splitting tensile strength three 150 mm × 300 mm cylindrical specimens were tested under a hydraulic jack from the lateral side, as per ASTM C496/C496M [43], the average of three tested specimens was considered.

### 4.4. Modulus of Elasticity

Moreover, 100 mm × 300 mm cylindrical samples were used to assess the modulus of elasticity according to ASTM C469/C469M [43]. Therefore, a steel ring with a strain gauge has been placed around the cylindrical sample and stress-strain data have been utilized to determine the initial and secant modulus of elasticity.

### 4.5. Water Absorption

After 28 curing days, water absorption was evaluated. The size of the sample shall not be less than 350 cm$^2$ and the specimens should be genuine without fractures or splitting on their surfaces, according to the standards defined by ASTM C642 [44].

### 4.6. Electric Resistivity Test

The electric resistivity test (ERT) was performed to evaluate the resistance and pore interconnections of concrete [45]. For this aim, three cylindrical specimens per mix, with a diameter of 100 mm and a height of 200 mm, were manufactured and tested under a voltage of 30 V at 28 days, as illustrated in Figure 4.

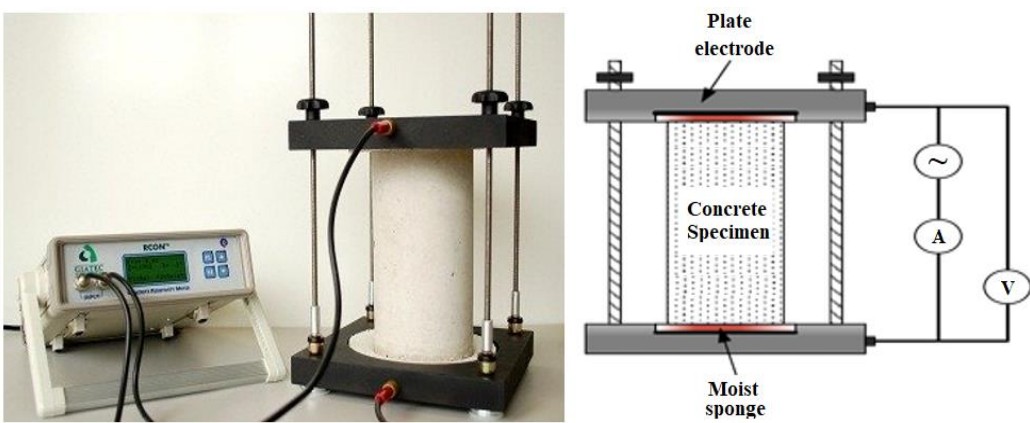

**Figure 4.** Test setup of the electric resistivity test.

## 5. Results and Discussion

### 5.1. Slump

The results of all concrete mixes are presented in Figure 5. According to Figure 5, increasing the value of NS and SP leads to decreasing the slump of concrete, and the impact of SP on that reduction is more significant than NS's. Therefore, the slump dropped by 3% and 6% by using 1% and 2% NS, respectively. Moreover, employing 5% and 10% SP reduces slump by about 4% and 9%, respectively, and using both 2% NS and 10% SP leads to a considerable reduction of about 14%. The previous investigations reported almost the same results thus confirm those presented in this study [46–48]. Jagadisha et al. [49] stated that adding NS unfavourably decreases the workability of concrete due to raising the viscosity of the mixes. The other previous studies showed a significant increase in the viscosity of concrete and a drop in slump value [50–52]. According to their results, the slump of concrete declined by 20% when 1.5% NS was used in the mixes. In another study, Duval and Kadri [53] performed a comprehensive evolution on the workability of concrete mixes produced with various SP contents. They showed that increasing SP led to reducing the slump value, so that they recommended increasing the superplasticizer in other to increase the slump.

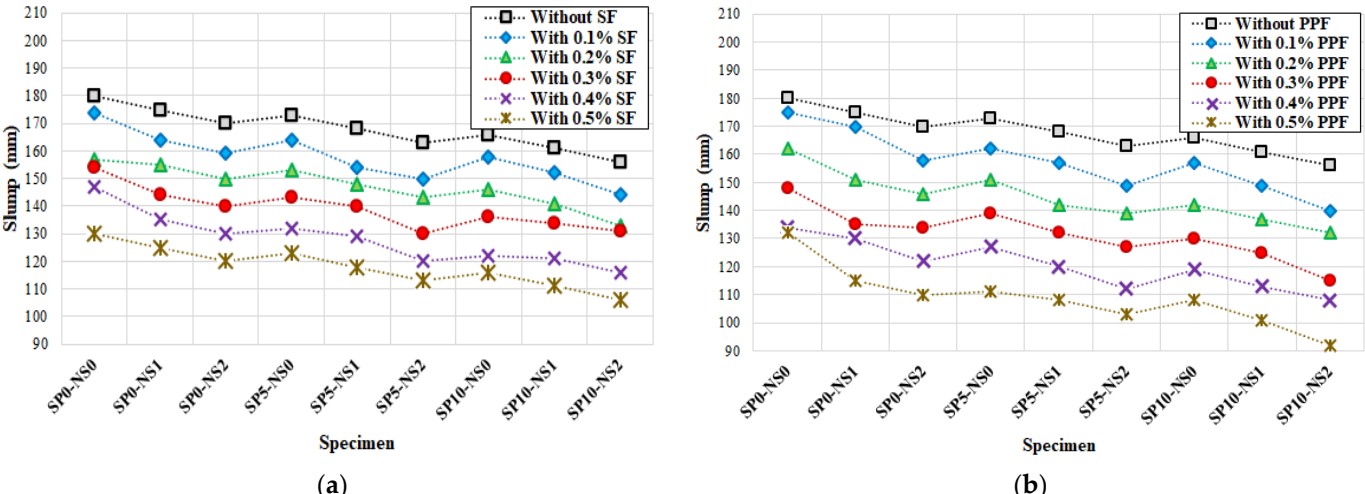

**Figure 5.** Effect of fibre on the slump of HSC containing NS and SP (**a**) SF and (**b**) PPF.

Additionally, increasing the SF content reduces the slump value (Figure 5). Besides, NS and SP have a negative effect on fibre reinforced HSC. Therefore, the slump dropped by approximately 6%, 11%, 17%, 22% and 28% when 0.1%, 0.2%, 0.3%, 0.4% and 0.5% SF were used, respectively. The reduction in slump due to using SF could be attributed to the bridging role of SF that keeps particles together and makes concrete paste denser [54,55]. Karimipour and Ghalehnovi [56] showed that the slump of concrete declined by 20% when 1% SF were added. In addition, using 2% NS and 10% SP together with 0.1%, 0.2%, 0.3%, 0.4% and 0.5% SF decreased the value of slump by about 19%, 24%, 30%, 36% and 41%, respectively. On the other hand, increasing the PPF content resulted in reducing the slump. Moreover, PPF reduces the slump flow diameter more than SF. This could be attributed to the higher water absorption of polymer fibres in comparison with SF [57]. In Figure 5, adding 0.1%, 0.2%, 0.3%, 0.4% and 0.5% PPF resulted in decreasing the slump by about 7.5%, 14%, 20%, 27% and 34%, respectively. Furthermore, using 2% NS and 10% SP with 0.1%, 0.2%, 0.3%, 0.4% and 0.5% PPF reduced the slump by roughly 20%, 27%, 33%, 40% and 47%, respectively. According to the results reported by Gencel et al. [58] the workability of concrete substantially declined when PPF were added due to increasing the contact between concrete paste and fibres.

The influence of SF and PPF on the workability of HSC can also be seen in Figure 6. According to this figure, using both SF and PPF have a substantial impact on dropping the

value of slump. To evaluate the impact of NS and SP on the workability of fibres reinforced HSC, the value of slump containing different NS and SP contents is illustrated in Figure 6.

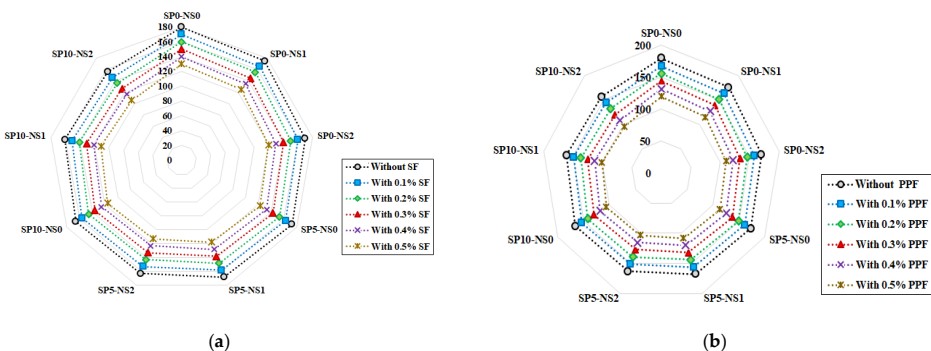

**Figure 6.** Impact of the fibres on the slump of HSC (**a**) SF and (**b**) PPF.

In SF reinforced HSC, a reduction trend occurred by raising the content of SN and SP. A similar tendency was observed in PPF reinforced HSC manufactured with NS and SP (Figure 7). By comparing Figure 7a,b, a greater impact of PPF on slump reduction is observed due to higher water demand and absorption of polymer fires (PPF in this study), in comparison with SF [32].

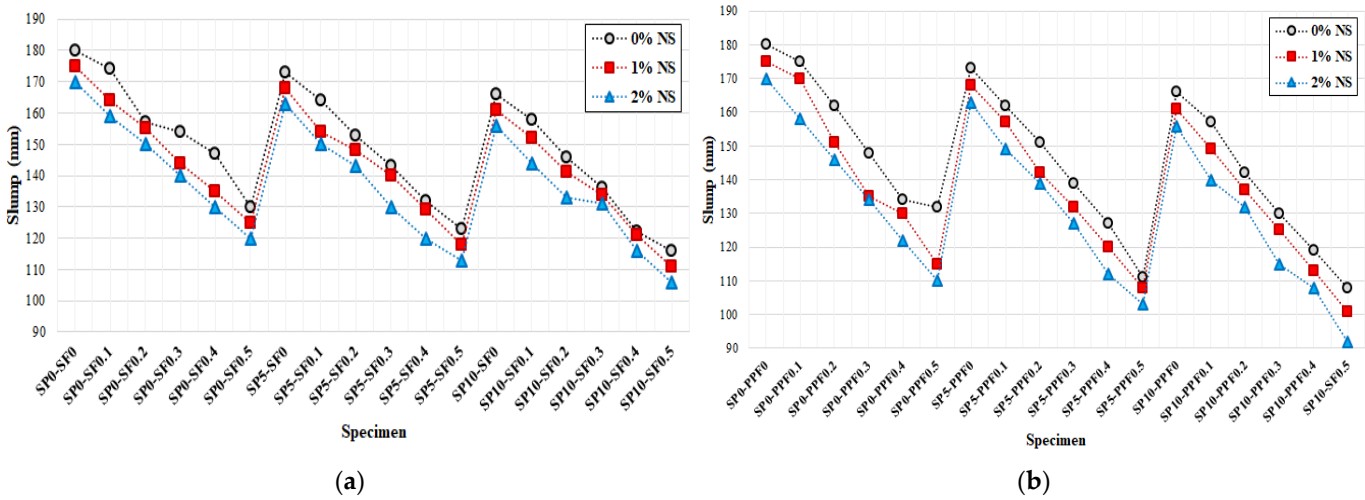

**Figure 7.** Influence of SP on the slump of HSC containing various NS and fibres (**a**) SF and (**b**) PPF.

### 5.2. Compressive Strength

Figure 8 shows that increasing the SF content improves the compressive strength of HSC. Duval and Kadri [49] showed almost the same results thus confirming those in the current study. They showed that pozzolanic activity of SP led to improving the compressive strength of concrete by up to 20%. In another investigation, Jalal et al. [52] found that, by using 2% NS in the mix, the compressive strength of concrete substantially improved. Li et al. [59] measured the mechanical characteristics of NS mortars and found that the compressive strength of those specimens was greater than that of the control sample with no NS. They stated that, if the NS is uniformly distributed in the cement paste, the microstructure of the concrete matrix could be improved, which results in improving the compressive strength. Besides, Khanzadi et al. [60] revealed that the compressive strength of concrete was improved by the use of NS, particularly at an early age.

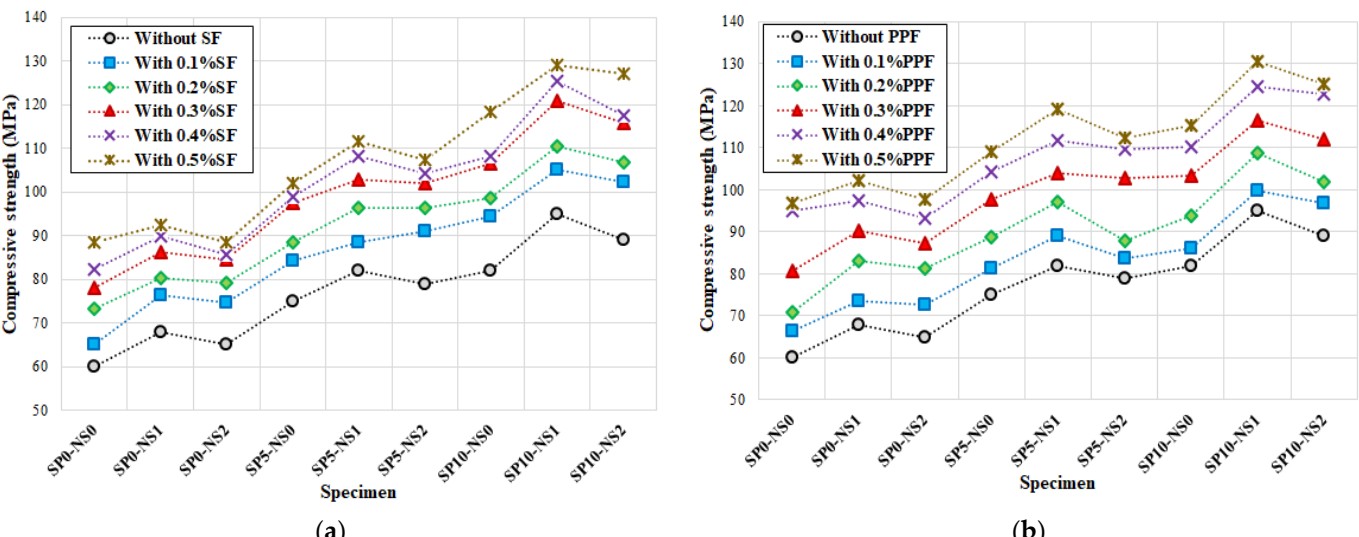

**Figure 8.** Influence of fibres on the compressive strength of HSC (**a**) SF and (**b**) PPF.

On the other hand, raising the amount of SP led to enhancing the compressive strength. Finally, increasing NS at up to 1% also enhances the compressive strength of HSC; however, using NS at more than 1% has a negative impact on this property. Furthermore, raising the SP content has a substantial influence on the improvement of HSC, especially when 10% SP is used. The compressive strength of HSC raised by almost 15%, 20%, 30%, 32% and 36% when 0.1%, 0.2%, 0.3%, 0.4% and 0.5% SF were employed, respectively. Moreover, adding 1% NS and 10% SP with 0.1%, 0.2%, 0.3%, 0.4% and 0.5% SF has the greatest impact on the HSC compressive strength: about 82%, 90%, 105%, 109% and 115%, respectively.

In addition, the improvement effect of SF on the compressive strength decreased by more than 0.3% SF when NS was employed. Additionally, the substantial positive impact of PPF on the compressive strength of HSC with NS and SP is seen in Figure 8. The main reason for improving the compressive strength of concrete with the use of fibres could be attributed to the bridging role of fibres, which create a strong core in the mid-height of the specimens. Karimipour and de Brito [61] performed a comprehensive evaluation of the influence of PPF on improving the compressive strength of concrete. They stated that, when a specimen is loaded by compression, a lateral expansion happens at the mid-height of the cylindrical sample. The bridging role of PPF improved the tensile strength of the matrix and enhanced the cohesion between concrete paste and aggregates and that limited lateral deformation of the sample, which enhanced the compressive behaviour of the specimen. Therefore, using 0.1%, 0.2%, 0.3%, 0.4% and 0.5% SF improved the compressive strength by about 15%, 22%, 34%, 43% and 50%, respectively. By comparing Figure 8a,b, increasing the value of SP did not have a negative effect on the improved performance of compressive strength when the value of PPF increased; however, using NS at more than 1% has a negative impact on this property. Besides, employing 1% NS and 10% SP together with 0.1%, 0.2%, 0.3%, 0.4% and 0.5% PPF enhances the compressive strength of HSC by approximately 66%, 81%, 99%, 113% and 123%, respectively.

In Figure 9, the effect of fibres on the compressive strength of HSC is illustrated in a different manner. As seen in Figure 9a, there is no substantial impact on this property between 0.1% and 0.2% SF. In addition, a similar trend is observed when 0.4% and 0.5% SF are used.

### 5.3. Splitting Tensile Strength

Figure 10 illustrates the splitting tensile strength of HSC containing various contents of fibres, NS and SP. The splitting tensile strength depends on the tensile strength of the matrix and the adhesion between cement and aggregates. As seen in Figure 10, increasing the value of SF resulted in raising the tensile strength of HSC. Using SP improves the splitting

tensile strength of HSC; however, the positive effect of SF on this property dropped by adding SP at more than 5%. Moreover, using 10% SP with no NS has a negative impact on the splitting tensile strength of HSC. On the other hand, using NS enhances the tensile strength of HSC. Almost the same improvement influence of SP and NS on the splitting tensile strength of concrete was reported by previous studies [10–13]. Moha and Hayat [62] showed that the splitting tensile strength of concrete substantially improved with the use of low SP content, while by increasing SP content, the splitting tensile strength gradually declined. The pozzolanic activity of SP is the main reason for improving the splitting tensile strength of HSC [63,64]. Moreover, the negative impact of using 10% SP dropped by raising the SF content. As seen in Figure 10, the tensile strength of HSC raised by nearly 15%, 20%, 26%, 30% and 37% by adding 0.1%, 0.2%, 0.3%, 0.4% and 0.5% SF, respectively. Besides, the tensile strength increased by about 65.5%, 86.4%, 88%, 92.6% and 104% when 0.1%, 0.2%, 0.3%, 0.4%, respectively, and 0.5% SF with 2% NS and 10% SP were employed. Duval and Kadri [53] showed that the SP particles size is an important property to improve the mechanical characteristics of concrete due to filling porous and creating a strong ITZ.

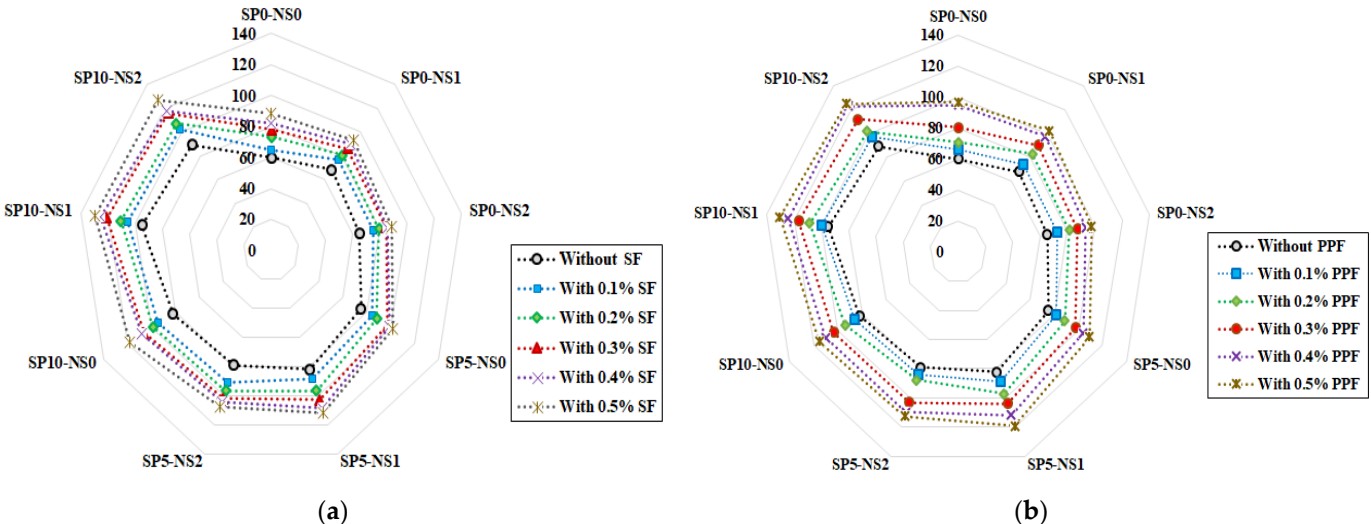

**Figure 9.** Impact of the fibres on the compressive strength of HSC containing NS and SP (**a**) SF and (**b**) PPF.

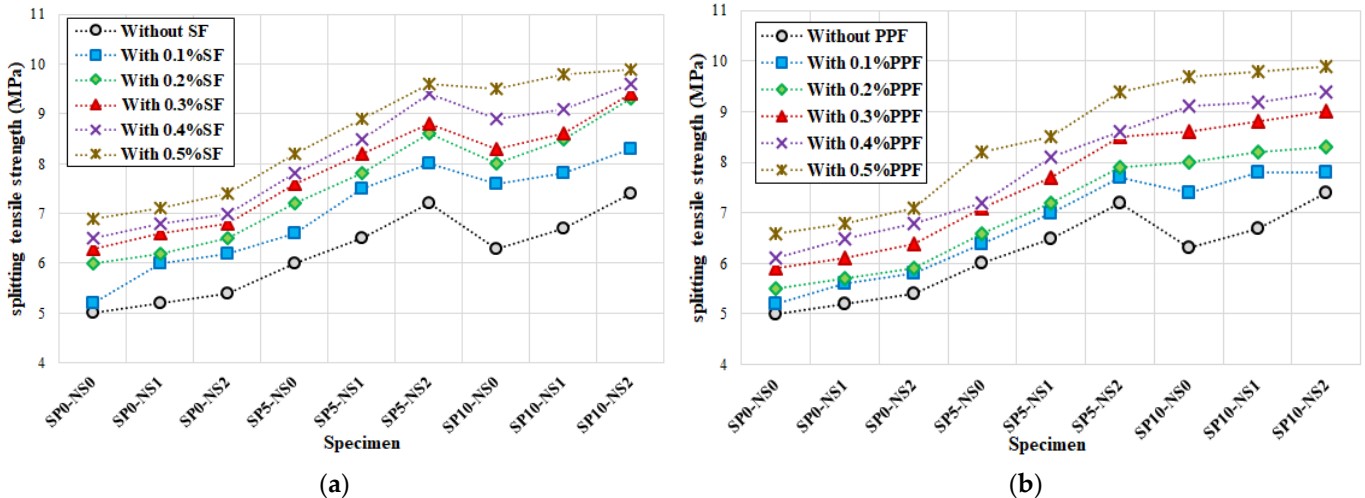

**Figure 10.** Influence of fibres of the splitting tensile strength of HSC (**a**) SF and (**b**) PPF.

In addition, using PPF resulted in improving the tensile strength of HSC. Furthermore, using PPF compensates for the negative influence of using SP at more than 5%. Furthermore, the impact of SF on the improvement of the tensile strength of HSC was greater than that

of PPF. The higher tensile strength of fibres, in comparison with concrete paste and the bridging role of fibres, are the main reasons why SF and PPF improve the splitting tensile strength of HSC [61]. Therefore, this property was improved by 7%, 10%, 18%, 25% and 31% by adding 0.1%, 0.2%, 0.3%, 0.4% and 0.5% PPF, respectively. Moreover, using both 2% NS and 10% SP with 0.1%, 0.2%, 0.3%, 0.4% and 0.5% PPF enhanced the splitting tensile strength of HSC by approximately 56.8%, 66.8%, 79.6%, 87% and 102%, respectively. Besides, Jalal et al. [52] stated that the use of low SP content improved the splitting tensile strength of concrete and it was further enhanced when SF was also added.

The effect of fibres on the splitting tensile strength of HSC can also be observed in Figure 11. Using either SF or PPF considerably improves the tensile strength of HSC.

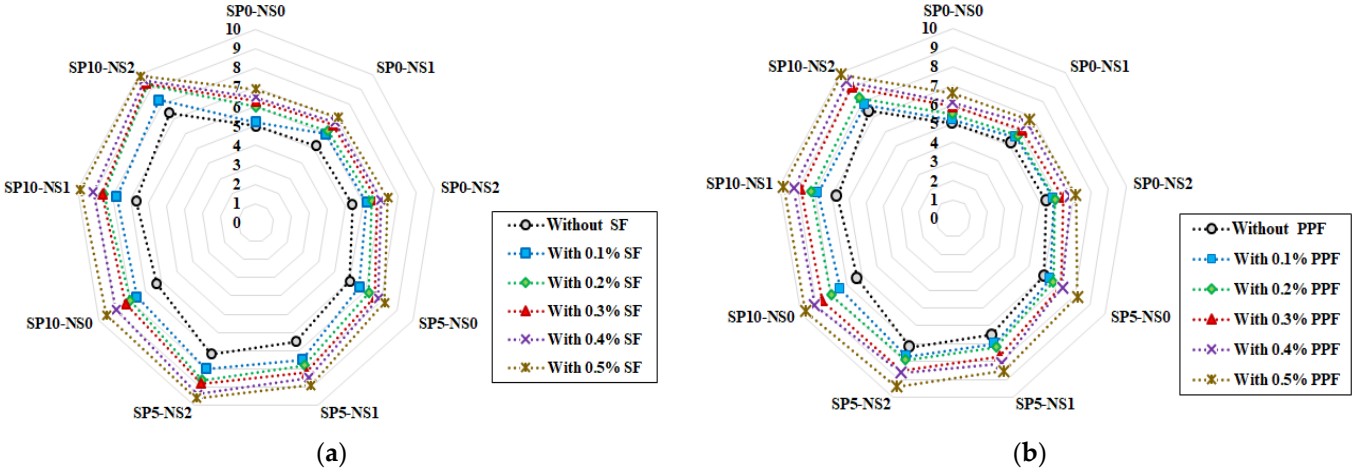

(a)             (b)

**Figure 11.** Effect of the fibres on the splitting tensile strength of HSC containing (**a**) NS and (**b**) SP.

### 5.4. Compressive-Splitting Tensile Strength Relationship

Figures 12 and 13 illustrate the correlation between the splitting tensile and compressive strengths of HSC with various contents of SF, PPF, NS and SP. Besides, formulas are proposed to consider the effect of SF and PPF. The proposed relationships (with $R^2$ above 0.83) are high-fitting with the experimental results.

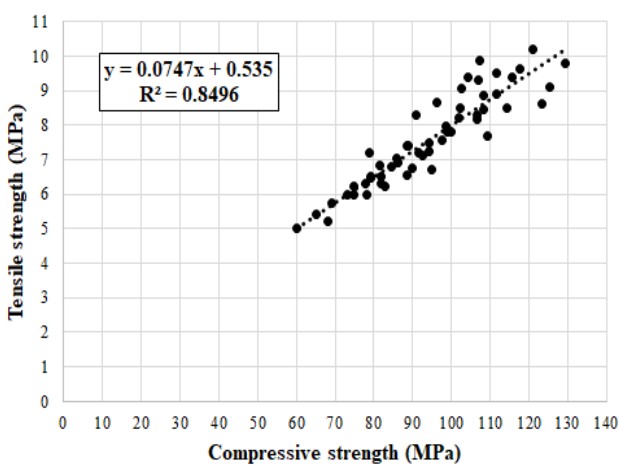

**Figure 12.** Compressive–tensile strength relationship of SF reinforced HSC containing NS and SP.

### 5.5. Modulus of Elasticity

The initial and secant moduli of elasticity of all concrete mixes are presented in Figures 14 and 15, where the influence of NS, SP, PPF and SP is illustrated.

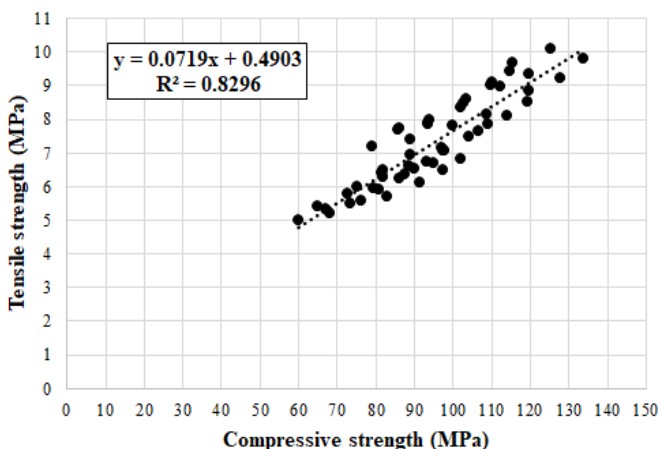

**Figure 13.** Compressive–tensile strength relationship of PPF reinforced HSC containing NS and SP.

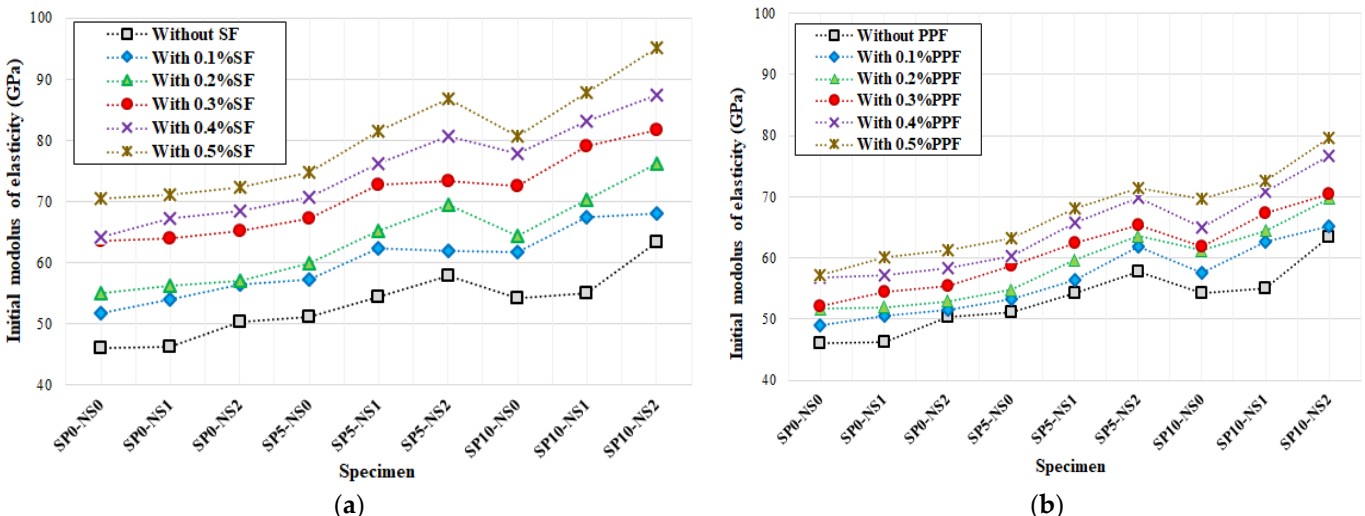

**Figure 14.** Effect of fibre on the initial modulus of elasticity of HSC containing NS and SP (**a**) SF and (**b**) PPF.

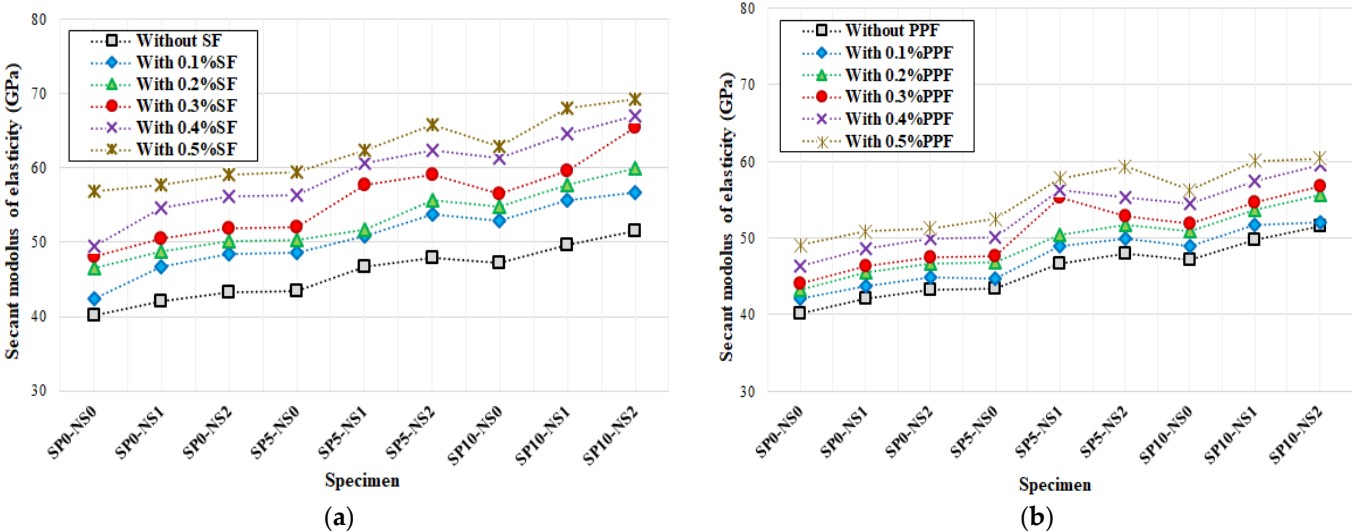

**Figure 15.** Effect of fibre on the secant modulus of elasticity of HSC containing NS and SP (**a**) SF and (**b**) PPF.

There, the modulus elasticity was raised by increasing the SF content as well as that of NS and SP. This could be attributed to the improvement of compressive strength due to the

bridging role of the fibres. Hannawi et al. [65] performed SEM of the interfacial transition zone (ITZ) and showed a strong crystal bond between concrete paste and aggregate when nanoparticles were incorporated. Besides, it showed that the modulus of elasticity of concrete improved by 6% and 11% when 1% and 1.5% NS was added. It also showed that the use of a large NS content adversely affected the modulus of elasticity of concrete. Also, the improvement effect of NS on the initial elasticity modulus raised by increasing the amount of SP There, this property improved by 15%, 20%, 36%, 42% and 50% when only 0.1%, 0.2%, 0.3%, 0.4% and 0.5% SF were added, respectively. Furthermore, using a combination of 2% NS and 10% SP with 0.1%, 0.2%, 0.3%, 0.4% and 0.5% SF led to enhancing this property by approximately 55%, 62%, 82%, 92% and 102%, respectively. On the other hand, a similar trend can be observed for the secant modulus of elasticity; naturally, the value of the secant modulus of elasticity is lower than that of the initial modulus of elasticity in each concrete mix.

Using PPF improves the value of the initial modulus of elasticity. Almost the same improvement in the modulus of elasticity due to PPF incorporation was reported by previous studies [61]. Małek et al. [66] measured the effect of PPF on the mechanical characteristics of conventional concrete. They showed that the use of 0.15% and 0.3% PPF improved the modulus of elasticity of plain concrete respectively by 15% and 20%, in comparison with the control sample. They also argued that PPF bridges the cracks and transfer stress over cracking regions. In addition, PPF creates a strong concrete core at the middles of cylindrical specimens, which led to increasing the stress resistance and reducing the lateral and axial strain of samples that led to improving the modulus of elasticity of fibres-reinforced HSC [61]. Moreover, employing NS and SP resulted in improving the modulus of elasticity. As seen in these figures, this property raised around 7%, 10%, 15%, 21% and 27% by using 0.1%, 0.2%, 0.3%, 0.4% and 0.5% PPF, respectively. Besides, incorporating 2% NS with 10% SP and 0.1%, 0.2%, 0.3%, 0.4% and 0.5% PPF led to enhancing this property by 44%, 48%, 55%, 63% and 71%, respectively. According to Table 8, the influence of SF on increasing the content of the moduli of elasticity of HSC is greater than that of PPF.

A high accuracy relationship between the moduli of elasticity and compressive strength was obtained, as illustrated in Figures 16 and 17. The proposed equations were established according to the type of fibres with R2 above 0.79 for the initial modulus of elasticity and 0.45 for the secant elasticity modulus. As seen in Figure 16, the slope of the relationship between the initial modulus of elasticity and compressive strength of PPF reinforced HSC was higher than that of SF reinforced HSC; however, an inverse trend is found for the secant modulus of elasticity (Figure 17).

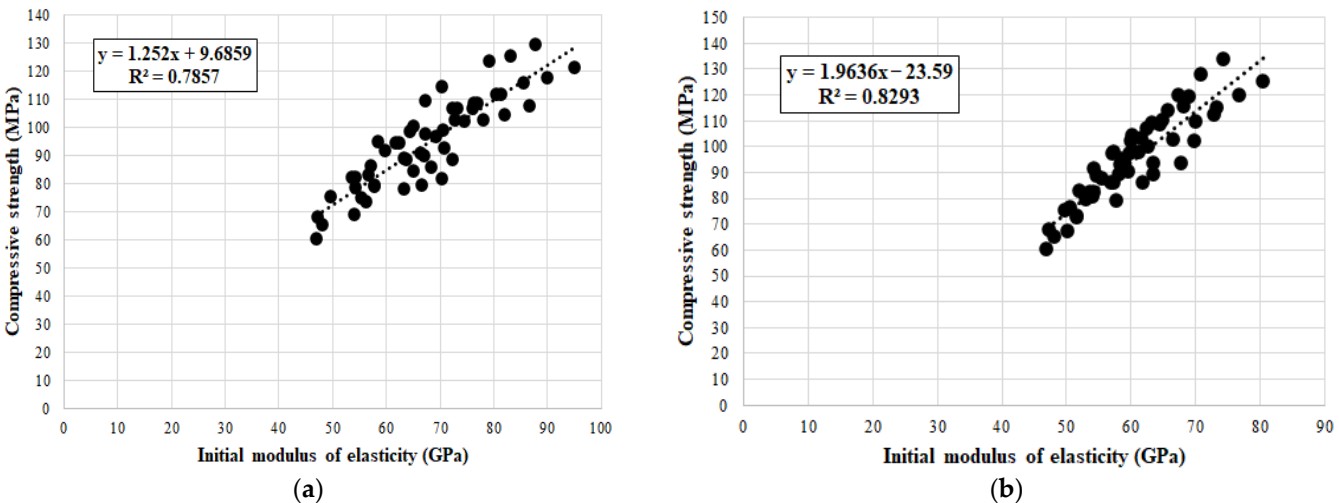

**Figure 16.** Effect of the fibres on the initial modulus of elasticity of HSC (**a**) SF and (**b**) PPF.

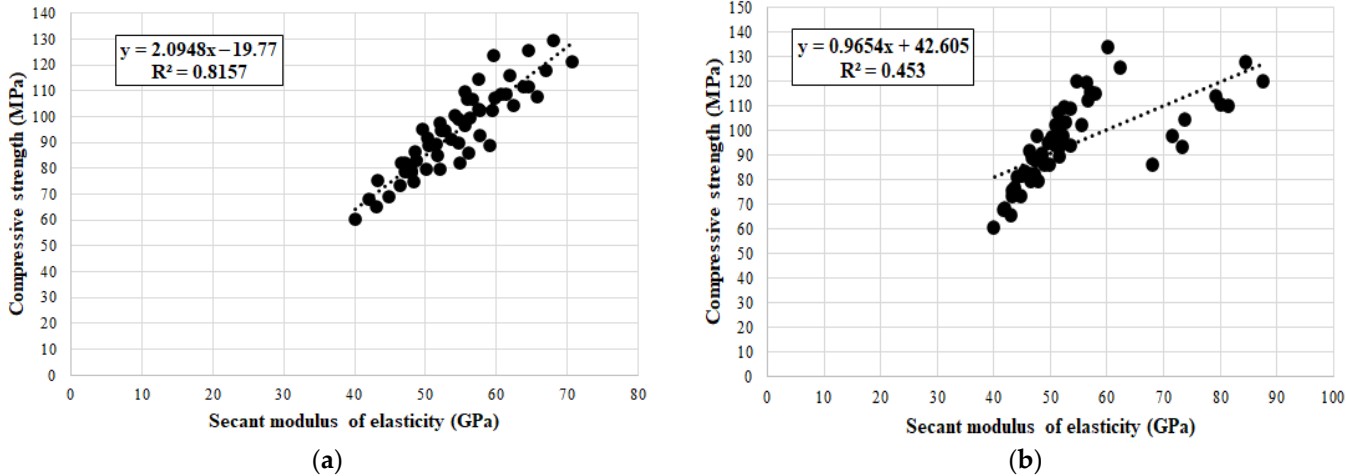

**Figure 17.** Effect of the fibres on the secant modulus of elasticity of HSC (**a**) SF and (**b**) PPF.

### 5.6. Water Absorption

The results of water absorption are represented in Figures 18 and 19. According to Figure 18, using SF resulted in raising the water absorption. Moreover, adding 10% SP significantly improved the water absorption of SF reinforced HSC. On the other hand, 5% SP reduces this property while using 1% NS considerably increased it when less than 5% SP was used. Using 1% NA dropped the water absorption when using 10% SP, and using a combination of 2% NS and 10% SP with 0.5% SF had a significant impact on increasing the water absorption. This could be attributed to the high surface area of NS. As shown by Mahdikhani et al. [67], the water absorption increased by adding NS, due to incremented specimens destruction, so that increasing the concrete specimens capillary porosity and permeability resulted by raising the acidity of the solution, while SP could act as a filler and reduce the porosity and permeability of HSC mixes. Additionally, Ali et al. [68] found that using a high SP content (20%) declined the workability by up to 20 mm. So, the incorporation of 0.5% SF and 10% SP with 0% and 2% NS raised the water absorption by about 56% and 68%, respectively. A similar trend was observed when PPF was used; however, the influence of PPF on the water absorption was more significant than that of SF. Therefore, using 10% SP and 0.5% PPF together with 0% and 2% NS increased this property by 70% and 82%, respectively (Figure 18b).

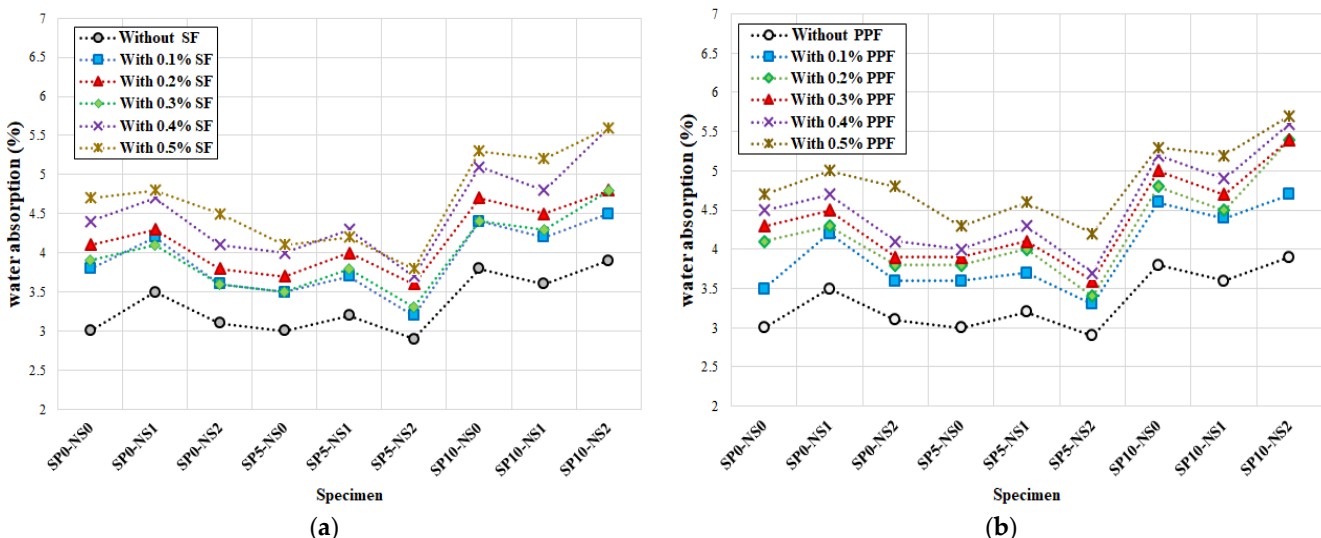

**Figure 18.** Impact of the fibres on the water absorption of HSC containing NS and SP (**a**) SF and (**b**) PPF.

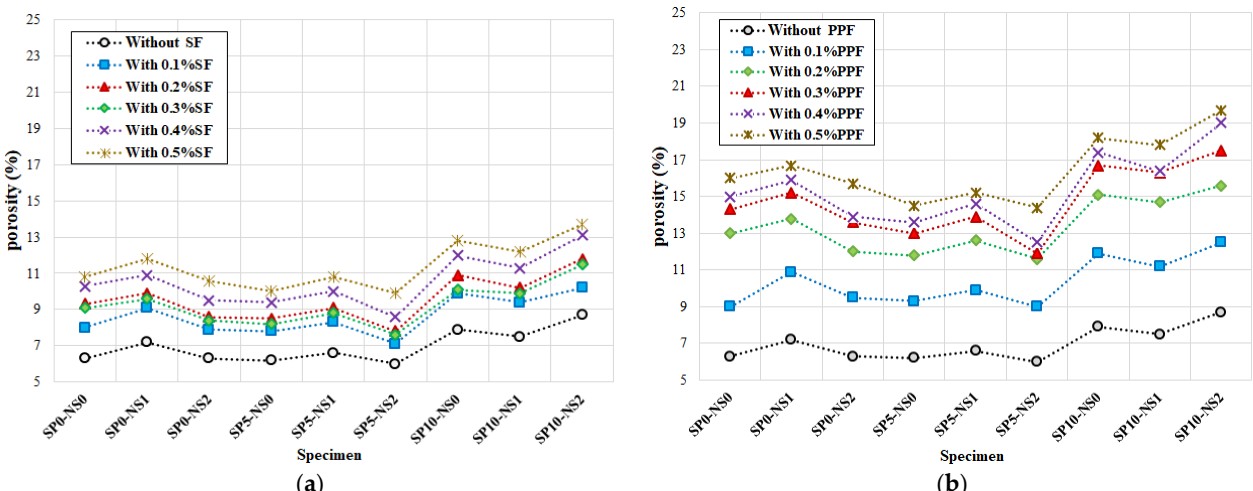

**Figure 19.** Effect of the fibres on the porosity of HSC containing NS and SP (**a**) SF and (**b**) PPF.

Figure 19 shows the impact of a combination of NS, SP, SF and PPF on the porosity of HSC. SF increased the water absorption of SF reinforced HSC, and further using 10% SP considerably it. Conversely, using 5% SP decreases the value of this property, whereas using 1% NS substantially enhanced it when less than 5% SP was employed. Besides, using 1% NA led to reducing the water absorption when using 10% SP and using 2% NS and 10% SP with 0.5% SF has a far-reaching influence on raising the water absorption. Therefore, a combination of 0.5% SF and 10% SP with 0% and 2% NS increased this property by nearly 103% and 115%, respectively. The same trend was experienced when PPF was used; however, the impact of PPF on water absorption was greater than that of SF. Consequently, using 0.5% PPF and 10% SP together with 0% and 2% NS raised this property by 147% and 173%, respectively (Figure 19b).

### 5.7. Electric Resistivity

Figure 20 shows the ERT results of specimens manufactured with different contents of NS, SP, SF and PPF. Increasing SF content resulted in a fall in ERT. Moreover, using 1% NS had a significant effect on improving ERT, and this property was enhanced by using NS at up to 2% while employing NS at more than 1% resulted in decreasing the ERT value. Alternatively, increasing the SP content did not have a significant effect on this property. Conversely, increasing the PPF content considerably improved the electric resistivity. Also, the ERT was significantly improved when PPF and 1% NS were combined.

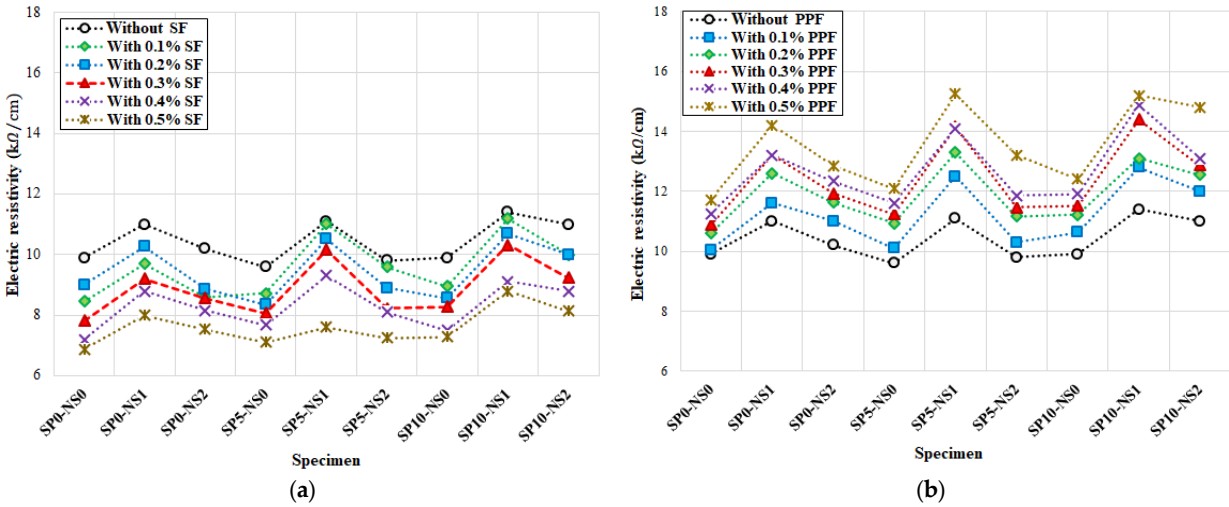

**Figure 20.** Influence of the fibres on the electrical resistivity of HSC containing NS and SP (**a**) SF and (**b**) PPF.

Therefore, this property was enhanced by about 57%, 62% and 68% when 0.5% SF and 1% NS were used with 0%, 5% and 10% SP, respectively. Previous investigations showed the same influence of PPF and SF on ERT thus confirming the results of our study. Recently, Karimipour [57] performed a study on the influence of different fibre types on the ERT. It showed that the main reason for reducing ERT results due to SF incorporation could be associated with the easier transfer of electricity across steel fibres while electricity transfer by polymer fibres is negligible.

## 6. Conclusions

In this study, 99 concrete mixes were designed and applied to assess the mechanical and durability performance of fibres reinforced high-strength concrete containing NS and SP. For this aim, SF and PPF were used at six-volume contents: 0%, 0.1%, 0.2%, 0.3%, 0.4% and 0.5%. Furthermore, NS were used at four contents: 0%, 1% and 2%. In addition, SP was used at three ratios by weight (0%, 5% and 10%). Therefore, the slump, compressive and splitting tensile strength, modulus of elasticity, water absorption and electric resistivity of the specimens were measured. Based on the experimental results, the following conclusions could be drawn:

1. Slump decreased by increasing the NS and SP contents. In addition, the influence of SP on decreasing the slump is greater than that of NS. In addition, adding fibres had a negative influence on the workability of HSC mixes and the reduction influence of PPF on the slump was greater than SF;
2. Adding SF and PPF improved the compressive and splitting tensile strengths of HSC and strengths were further enhanced with increasing fibres contents. Conversely, the optimal content of NS and SP was obtained by 1% NS and 10% SP in terms of maximum compressive and splitting tensile strengths;
3. The initial modulus of elasticity is raised by increasing the SF content, as well as that of NS and SP. Additionally, using PPF improved the initial modulus of elasticity. Using NS and SP resulted in raising the modulus of elasticity by around 7%, 10%, 15%, 21% and 27% by using 0.1%, 0.2%, 0.3%, 0.4% and 0.5% PPF, respectively;
4. Using SF increased the water absorption. Adding 10% SP significantly increased the water absorption of SF reinforced HSC. Oppositely, 5% SP reduced the value of this property while using 1% NS increased water absorption considerably when less than 5% SP was used;
5. Using 1% NS had a substantial impact on ERT results: this property was enhanced by using NS at up to 2% while employing it at more than 1% resulted in a drop. On the other hand, increasing the value of SP did not have a significant influence on this property. Increasing PPF content considerably enhanced the electric resistivity. ERT results were also significantly improved when both PPF and 1% NS were used.

**Author Contributions:** Conceptualization, M.G. and J.d.B.; Formal analysis, M.G. and M.E. Investigation, A.K. and M.E.; Methodology, A.K. and M.G.; Software, A.K. and M.E.; Supervision, J.d.B.; Writing—original draft, A.K., M.G. and M.E.; Writing—review and editing, J.d.B. All authors have read and agreed to the published version of the manuscript.

**Funding:** This research received no external funding.

**Institutional Review Board Statement:** Not applicable.

**Informed Consent Statement:** Not applicable.

**Data Availability Statement:** The raw/processed data required to reproduce these findings cannot be shared at this time as the data also forms part of an ongoing study.

**Acknowledgments:** The authors gratefully acknowledge the support of CERIS (Civil Engineering Research and In-novation for Sustainability) and FCT (Foundation for Science and Technology).

**Conflicts of Interest:** The authors declare that they have no conflict of interest.

## Abbreviations

ERT  electric resistivity test
HSC  high-strength concrete
NS  nano-silica
PPF  polypropylene fibres
SCC  self-compacting concrete
SF  steel fibres
SP  silica fume powder

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
