# Peer review of "Properties of Fibre-Reinforced High-Strength Concrete with Nano-Silica and Silica Fume"

_applsci, doi:10.3390/app11209696_

Round 1

Reviewer 1 Report

The article, although it contains numerous and interesting data and has a complete and exhaustive introduction, should be rewritten giving explanations on the results obtained. In the current state more than a scientific article it seems a laboratory report of the results obtained. A part of discussion of the results obtained and a thorough comparison with what has been previously obtained in the literature is completely missing. In its current state it cannot be accepted.
Minor problems: superscripts and subscripts of chemical formulas must be revised, the percentage by volume is not indicated if compared to the quantity of cement used, the total volume of the sample or other.

Author Response

RESPONSES TO Reviewer #1’s COMMENTS:

The authors would like to express their great appreciation towards you for your thorough and detailed review of our manuscript. Without a doubt, the presented ideas and the additional recommended actions have strengthened the article. Almost all of your comments have been taken into account in the paper. The places in the text where these suggestions are considered are marked in Red. The answers to all of your valuable comments are given in the following lines.

  1. COMMENT:The article, although it contains numerous and interesting data and has a complete and exhaustive introduction, should be rewritten giving explanations on the results obtained. In the current state more than a scientific article it seems a laboratory report of the results obtained. A part of a discussion of the results obtained and a thorough comparison with what has been previously obtained in the literature is completely missing.

RESPONSE: This comment was taken into account in the revised file and a discussion about the previous studies have been carried out for each test in the section of “Results and Discussion” to confirm the accuracy of the presented results in the current study.

  1. COMMENT: superscripts and subscripts of chemical formulas must be revised, the percentage by volume is not indicated if compared to the quantity of cement used, the total volume of the sample or others.

RESPONSE: All superscripts and subscripts of chemical formulas have been revised as well as their volume.

Again, we appreciate all of your insightful comments. We worked hard to be responsive to them. Thank you for taking the time and energy to help us improve the paper.

Reviewer 2 Report

Dear Authors,

Thank you for your manuscript, you have a lot of data and a very poor choice of how to reflect your obtained results in charts. Please provide your data information in charts (not both charts and tables for the same data), excel column chart is not the best way to reflect your data.

Please choose gradation to show in table or figure.

Can you please organize you tables in a different compact way? Maybe just allocate info in rows and not in columns? It doesn’t really look good now

Pleas provide a short comprehensive information about your 99 mixes, split them in groups and give short description why those tested and what is difference between them. Why all of them tested, could not be possible to reduce amount of tested mixes. Explain the naming of your samples!

Slump of mixes – please provide just a chart with values – again you can create groups for a better overview of your results. Please provide info on your charts to understand where you have NS where SP. I suggest to make a better naming of your samples. Since you have so much data, please consider to make it as more easy visualized as better for a reader.

Table 7, Figures 5-7 – please don’t repeat the same info and in table and in figures, just choose one – easy comprehensive on watching at it from the first glance

Combine figures 8-9-10, not a good choice of chart

You need to rethink how you explain your results for so many samples, all these % indication and comparison is tiring. Line 236: doubled %%

Can you clearly indicate how many samples you test per mix? One?

Why in Figure 11 you have 57 samples and in Figure 12  you have 104 samples?

Why data provided in table 8, why not just chart?

Discussion part is missing! Please add one, indicate what is novel in your study and refer to some references

Conclusions shorten!

Author Response

RESPONSES TO Reviewer #2’s COMMENTS:

The authors would like to express their great appreciation towards you for your thorough and detailed review of our manuscript. Without a doubt, the presented ideas and the additional recommended actions have strengthened the article. Almost all of your comments have been taken into account in the paper. The places in the text where these suggestions are considered are marked in Green. The answers to all of your comments are given in the following lines.

  1. COMMENT: Please provide your data information in charts (not both charts and tables for the same data), an excel column chart is not the best way to reflect your data.

RESPONSE: This comment was taken into account and data information was provided in the chart style. Also, a few inappropriate column charts were replaced with appropriate charts. Due to a large number of test results, the authors think that the best chart is column style for some results since different chart styles were checked by the authors.

  1. COMMENT: Please choose gradation to show in table or figure.

RESPONSE: The gradation was shown in a figure.

  1. COMMENT:Can you please organize your tables in a different compact way? Maybe just allocate info in rows and not in columns? It doesn’t really look good now.

RESPONSE: Most of the tables have been eliminated and the results are presented in charts.

  1. COMMENT:Please provide a short comprehensive information about your 99 mixes, split them into groups and give a short description of why those were tested and what is difference between them is. Why all of them were tested, could not be possible to reduce the number of tested mixes. Explain the naming of your samples

RESPONSE: This comment was taken into account in the revised file. The name of specimens was modified and described in the revised file. In addition, the concrete mixes were split into different groups based on used materials. A wide range of specimens was produced and tested in this study to be sure about the mechanical performance of HSC. Additionally, the authors believe that the results presented in this study could be used as a good database for future studies. Therefore, removing some results would lead to creating a gap in this study.

  1. COMMENT: Slump of mixes – please provide just a chart with values – again you can create groups for a better overview of your results. Please provide info on your charts to understand where you have NS where SP. I suggest making a better naming of your samples. Since you have so much data, please consider making it as easier visualized as better for a reader.

RESPONSE: This comment was taken into account and the slump results have been presented in chart format with an appropriate shape. Also, the name of the specimens was modified.

  1. COMMENT: Table 7, Figures 5-7 – please don’t repeat the same info and in the table and figures, just choose one – easy comprehensive on watching at it from the first glance

RESPONSE: This comment was taken into account and the results are only illustrated in charts.

  1. COMMENT: Combine figures 8-9-10, not a good choice of chart.

RESPONSE: This comment was taken into account and thefigures were merged.

  1. COMMENT: You need to rethink how you explain your results for so many samples, all these % indication and comparison is tiring. Line 236: doubled %%

RESPONSE: This comment was taken into account in the revised file.

  1. COMMENT: Can you indicate how many samples you test per mix? One?

RESPONSE: From each mix, three specimens were tested and the average of three tested specimen was considered.

  1. COMMENT: Why in Figure 11 you have 57 samples and in Figure 12 you have 104 samples?

RESPONSE: The number in Fig. 12 was wrong and this figure was modified based on your previous comments.

  1. COMMENT: Why is data provided in table 8, why not just chart?

RESPONSE: The results were presented in both chart and table styles for better understanding by readers. But, according to your first comment, the results are only presented in chart style and charts are modified, as well.

  1. COMMENT: The discussion part is missing! Please add one, indicate what is novel in your study and refer to some references

RESPONSE: This comment was taken into account in the revised file. The novelty of this study was explained in the “Research Significance” section. In addition, a discussion about the previous studies has been carried out for each test in the section of “Results and Discussion”.

  1. COMMENT: Conclusions shorten!

RESPONSE: The conclusion section was shortened.

Again, we appreciate all of your insightful comments. We worked hard to be responsive to them. Thank you for taking the time and energy to help us improve the paper.

Round 2

Reviewer 1 Report

The author addressed all the issue, the article is well written.

Author Response

Thank you for your positive view about the paper

Reviewer 2 Report

Dear Authors,

Your paper looks better now but still some work to do:

  1. please choose the same style for all your plots (font style and size, color, italic or bold, outline – black or transparent, etc.) otherwise you have quite a variety (it is, a scientific paper, not a presentation, therefore please follow formatting in the same style for all charts)
  2. Tables 1, 2, 3 and 5 – reorganize, make them more compact and better for overview.
  3. Lines 187-188: is mentioned that you have 1782 specimens. Can you provide an overview in table of performed tests and amount of samples? Still not clear how many samples you have used to test per one mix. Please indicate it in your manuscript.
  4. Methods part (the tests performed and according to which standards) should be extracted from Results and Discussion section. Please follow the standard outline: Introduction, Materials, Methods, Results, Discussion and Conclusions.
  5. The conclusions part is too long. Please shorten and polish your wording. This part requires more focus and work on it!

Author Response

RESPONSES TO Reviewer #2’s COMMENTS:

The authors would like to express their great appreciation towards you for your thorough and detailed review of our manuscript. Without a doubt, the presented ideas and the additional recommended actions have strengthened the article. Almost all of your comments have been taken into account in the paper. The places in the text where these suggestions are considered are marked in Red. The answers to all of your comments are given in the following lines.

  1. COMMENT: please choose the same style for all your plots (font style and size, color, italic or bold, outline – black or transparent, etc.) otherwise you have quite a variety (it is, a scientific paper, not a presentation, therefore please follow formatting in the same style for all charts).

RESPONSE: Done as requested.

  1. COMMENT: Tables 1, 2, 3 and 5 – reorganize, make them more compact and better for overview.

RESPONSE: The mentioned tables were re-organized.

  1. COMMENT:Lines 187-188: is mentioned that you have 1782 specimens. Can you provide an overview in table of performed tests and amount of samples? Still not clear how many samples you have used to test per one mix. Please indicate it in your manuscript.

RESPONSE: This comment is clearly mentioned in the text. In table 1, 99 mixes were presented. Also, six tests were carried out: slump, compressive and splitting tensile strengths, modulus of elasticity, water absorption and electrical resistivity. In each test, three specimens were produced for each mix and the average of three samples was considered. Therefore, 99 ⨯ 6 ⨯ 3 is equal to 1782.

  1. COMMENT:Methods part (the tests performed and according to which standards) should be extracted from Results and Discussion section. Please follow the standard outline: Introduction, Materials, Methods, Results, Discussion and Conclusions

RESPONSE: The paper was reorganized in your mentioned outline: Introduction, research significance, materials, methods, results and discussion and conclusions.

  1. COMMENT: The conclusions part is too long. Please shorten and polish your wording. This part requires more focus and work on it!

RESPONSE: The conclusion section was shortened as much as possible.

Again, we appreciate all of your insightful comments. We worked hard to be responsive to them. Thank you for taking the time and energy to help us improve the paper.
